# MAVS-Based Reporter Systems for Real-Time Imaging of EV71 Infection and Antiviral Testing

**DOI:** 10.3390/v15051064

**Published:** 2023-04-26

**Authors:** Xiaozhen Li, E Yang, Xinyu Li, Tingting Fan, Shangrui Guo, Hang Yang, Bo Wu, Hongliang Wang

**Affiliations:** 1Department of Pathogen Biology and Immunology, Xi’an Jiaotong University Health Science Center, Xi’an 710061, China; 2Key Laboratory of Environment and Genes Related to Diseases, Xi’an Jiaotong University, Xi’an 710061, China

**Keywords:** enterovirus, reporter, fluorescence, live cell imaging, antiviral

## Abstract

Enterovirus consists of a variety of viruses that could cause a wide range of illness in human. The pathogenesis of these viruses remains incompletely understood and no specific treatment is available. Better methods to study enterovirus infection in live cells will help us better understand the pathogenesis of these viruses and might contribute to antiviral development. Here in this study, we developed fluorescent cell-based reporter systems that allow sensitive distinction of individual cells infected with enterovirus 71 (EV71). More importantly, these systems could be easily used for live-cell imaging by monitoring viral-induced fluorescence translocation after EV71 infection. We further demonstrated that these reporter systems could be used to study other enterovirus-mediated MAVS cleavage and they are sensitive for antiviral activity testing. Therefore, integration of these reporters with modern image-based analysis has the potential to generate new insights into enterovirus infection and facilitate antiviral development.

## 1. Introduction

Enteroviruses are a group of positive-sense single-stranded RNA viruses belonging to the picornaviridae family. Based on the molecular and antigenic properties, they are divided into 15 species, seven of which can infect humans, including four enteroviruses (EV-A, EV-B, EV-C, EV-D) and three rhinoviruses (RV-A, RV-B, RV-C) [1,2]. Enteroviruses can cause a wide range of clinical illnesses in humans. Although most of them were reported to be mild and self-limited, there has been a recent increase in the incidence of enterovirus-induced morbidity and serious diseases, including myocarditis, acute flaccid myelitis, encephalitis, pancreatitis and hepatitis [3,4,5,6]. It has become a huge public health burden globally, especially in the Asia-Pacific region [7,8]. However, there are still no specific therapeutic drugs for enteroviruses.

The genome of enteroviruses ranges from 7100 to 8500 nucleotides, and contains a large open reading frame (ORF) flanked by a 5′-untranslated region (5′-UTR) and a 3′-untranslated region (3′-UTR) [9]. The large ORF encodes a polyprotein, which is processed into four viral structure proteins (VP1, VP2, VP3, VP4) and seven nonstructural proteins (2A, 2B, 2C, 3A, 3B, 3C, 3D) [10]. These non-structural proteins play crucial roles in the replication and translation of enteroviruses [11].

Innate immunity is the first line of host defense against viral invasion. During enterovirus infection, the viral components and replication intermediates activate retinoic acid-inducible gene I (RIG-1)-like receptor and MDA5, which will interact with the mitochondrial antiviral-signaling protein (MAVS, also known as IPS1) to activate immune response to inhibit viral replication and transmission [12,13,14]. On the other hand, viruses have evolved different mechanisms to escape innate immunity, for example, by cleaving the MAVS protein [15]. MAVS is a 540-amino acid protein containing three domains: an N-terminal caspase recruitment domain (CARD), an intermediate proline-rich region (PRR), and a C-terminal transmembrane (TM) domain, and it is localized primarily to the outer mitochondrial membrane [16]. A variety of viruses have been reported to cleave MAVS with their viral-encoded proteases, including hepatitis C virus (HCV), hepatitis A virus(HAV), human rhinovirus type C, coxsackievirus B3 (CVB3), etc. [17,18,19,20,21]. Different viruses have been shown to cleave MAVS at different sites, for example, HCV cleaves MAVS at the Cys508, while enterovirus A71 (EV71) cleaves MAVS at Gly209, Gly251 and Gly265 to block type I IFN responses [22].

In this study, we developed fluorescent reporter systems to identify enterovirus infected cells by taking advantage of the viral protease-mediated MAVS cleavage and fluorescence relocalization. Through monitoring the nuclear translocation of RFP or relocalization of GFP, these systems enable quick and sensitive distinction of enterovirus infected cells by fluorescent microscopy. We further demonstrated that three different enteroviruses-EV71, Echovirus 7 (Echo7) and coxsackievirus B5 (CVB5) all could cleave MAVS at the same residues, while HCV cleaves MAVS at a distinct site. More importantly, these systems could be used to monitor live-cell infections and provide a new method for antiviral testing. Therefore, integration of these fluorescent translocation features with other cellular phenotypes has the potential to creating new platforms for basic enterovirus studies and antiviral drug development.

## 2. Materials and Methods

### 2.1. Cells and Viruses

Human embryonic kidney 293T cells, human rhabdomyosarcoma RD cells, and human hepatoma Huh7 cells were obtained from the American Type Culture Collection (ATCC). These cells were grown at 37 °C and 5% CO_2_ in Dulbecco’s modification of Eagle’s medium (DMEM, Thermo Scientific, Waltham, MA, USA) supplemented with 10% fetal bovine serum, 100 units/mL penicillin G, and 100 μg/mL streptomycin sulfate (Beyotime Biotechnology, Shanghai, China). Huh-7 cells stably expressing the reporters were cultured in complete medium containing 4 μg/mL blasticidin (InvivoGen, San Diego, CA, USA).

The EV71, Echo7 and CVB5 strains used in the study were obtained from the Xi’an CDC and has been previously described [23,24]. They were amplified and titrated in RD cells. The HCV-JFH1 strain (genotype 2a) was a gift of Dr. Takaji Wakita (National Institute of Infectious Diseases, Tokyo, Japan).

### 2.2. Antibodies and Reagents

Mouse monoclonal antibodies against MAVS was purchased from Proteintech (Wuhan, China). Rabbit polyclonal antibody against RFP, Mouse monoclonal antibodies against Enterovirus, AlexaFluor-488 and AlexaFluor-555 conjugated goat anti-mouse IgG, Hoechst 33342 and DAPI were purchased from Thermo Scientific (Waltham, MA, USA). Rabbit monoclonal antibody against eGFP was purchased from Zenbio Science (Chengdu, China). Rabbit monoclonal antibody against β-Actin was from ABclonal Technology (Woburn, MA, USA). GuHCl was purchased from Sigma-Aldrich (St. Louis, MO, USA).

### 2.3. MAVS-Based Reporter Cell Lines Generation

The coding sequence of MAVS was amplified by PCR from Huh-7 cDNA library and subcloned into a lentiviral backbone (pHW200) together with tagRFP and SV40 nuclear localization signal (NLS, PKKKRKVG) or eGFP tag to generate RFP-NLS-MAVS or GFP-MAVS. MAVS truncations or mutations were generated by Gibson Assembly with 2× MultiF Seamless Assembly Mix (ABclonal, Wuhan, China) with primers specified in Appendix A. All constructs were confirmed by sequencing.

MAVS constructs were then cotransfected into 293T cells together with psPAX2, pMD2.G to prepare lentivirus as described previously [25]. Huh 7 cells were then transduced with lentivirus and stable cell lines were generated with blastcidin selection.

### 2.4. Microscopy and Image Taking

Cells were seeded on poly-D-lysine slides and were either mock infected or infected with indicated enterovirus for 16 h before they were fixed with ice cold methanol for 15 min. Cells were then blocked with 3% BSA for 1 h at room temperature, followed by MAVS, VP3 antibody staining and corresponding Alexa Fluor 488/555-conjugated secondary antibody labeling. Images were taken with FV3000 confocal microscope (Olympus, Tokyo, Japan) after samples were mounted with ProLong Antifade reagent (Thermo Scientific, Waltham, MA, USA).

For time-lapse cell imaging, cells were seeded in Incucyte^®^ Imagelock 96-well plates for 24 h followed by viral infection. Cells were then placed in the Cytation 5 Cell Imaging Multi-Mode Reader in combination with the BioSpa 8 Auto Incubator (BioTek Instruments, Santa Clara, CA, USA). Images were taken every 2.5 h for a total of 30 h to monitor the real-time live cell fluorescence imaging. All images were processed with NIH Fiji/ImageJ.

### 2.5. Western Blot Analysis

Cells were treated as indicated and harvested with LDS Sample buffer (Thermo Scientific, Waltham, MA, USA) and then separated with 10% SDS-PAGE. Proteins were transferred onto PVDF membranes and then blocked for 1 h at room temperature in 5% non-fat milk followed by primary antibody and secondary antibody labeling. Blots were visualized were using enhanced chemiluminescence reagent (Mishushengwu, Xi’an, China) with FUSION SOLO 6S (Vilber, Paris, France).

### 2.6. Viral RNA Quantification by RT-qPCR

The reporter cells were first infected with EV71 for 4 h followed by serial dilutions of GuHCL treatment for another 20 h. Cells were then harvested and total cellular RNA was extracted using Cell Total RNA isolation kit (FOREGENE, Chengdu, China) according to manufacturer’s instruction. cDNA was reverse transcribed with qPCR RT kit (NEB, Ipswich, MA, USA) and qPCR was performed with SYBR Green PCR mix (GenStar, Beijing, China) using the following primers: EV71-F: 5′ CAAGGTTCCAGCACTCCAAG 3’; EV71-R: 5’CCGCCCTACTGAAGAAACTATC 3′. A pTopo-EV71 plasmid was serially diluted to generate a standard curve to calculate the viral genome copy numbers.

### 2.7. Statistical Analysis

Prism 8.4.3 software was used for statistical analyses and nonlinear regression curve fitting to calculate IC_50_ and CC_50_ values. P values less than 0.05 were considered statistically significant. All values are depicted as mean ± standard deviation (SD). At least three replicates were performed for each experiment.

## 3. Results

### 3.1. MAVS-Based Reporter Systems for Detection of EV 71 Infection

Based on the feature that EV71 could cleave MAVS at amino acid sites Gly209, Gly251, and Gly265 with its 2A protein, cell-based fluorescent reporting systems were constructed to visualize EV71 infection. We first established a reporter system by fusing tagRFP and a truncated form of MAVS (aa 201–272/509–540) along with a nuclear localization signal (RFP-NLS-MAVS2). EV71 infection and the expression of 2A protein will lead to cleavage of MAVS, resulting in the translocation of the tagRFP from puncta mitochondria staining into nucleus (Figure 1A, left panel). Similarly, a GFP version of this reporter system was also constructed by fusion GFP with truncated MAVS. Of note, no NLS was included in this GFP construct (GFP-MAVS2), therefore, viral infection will only release the GFP from the mitochondria and result in diffused GFP expression (Figure 1A, right panel). Huh 7 cells stably expressing RFP-NLS-MAVS2 or GFP-MAVS2 were first stained with MAVS antibody and then observed under confocal microscope. Figure 1B showed that both forms of MAVS showed puncta signals featuring mitochondria staining before infection and either tagRFP or GFP showed perfect colocalization with MAVS staining, suggesting that fusion expression of fluorescence protein does not change the distribution and localization of MAVS. When these cells were infected with EV71 for 20 h, tagRFP was found to translocate into nucleus and colocalized with Hoechst, while GFP showed diffused distribution (Figure 1C), suggesting the successful cleavage of MAVS by viral proteins. Viral protein expression was confirmed by immunostaining (Appendix A), which supported the correspondence between fluorescence relocalization and viral protein expression at single cell levels. These results suggested that these fluorescence reporter systems are suitable to detect EV71 infections.

To confirm the specificity of cleavage site of MAVS by EV71 virus, another truncated form of MAVS (aa 462–540) was fused with tagRFP (RFP-NLS-MAVS4) and tested for its expression pattern. HCV has been reported to cleave MAVS at aa508, and was shown to translocate this cleaved RFP into nucleus (Appendix A), however, EV71 infection could not, despite the successful expression of viral proteins (Appendix A). On the contrary, MAVS2 could be cleaved by EV71 but not by HCV (Appendix A). Taken together, these results suggest that the translocation of tagRFP into nucleus was caused by viral protease cleavage at specific sites of MAVS molecules. Finally, EV71 could induce cell death after infection and to exclude the possibility that the translocation of RFP was nonspecifically caused by cell death and pyknosis, we treated Huh 7 cells stably expressing RFP-NLS-MAVS2 with puromycin and found that although cytotoxicity was observed after treatment, no apparently colocalization of RFP with Hoechst was observed (Appendix A), suggesting that the nucleus translocation that we observed was specifically caused by EV71 cleavage of MAVS.

### 3.2. RFP-NLS-MAVS Reporter Could Be Cleaved by Different Enteroviruses

As described above, EV71 is known to cleave MAVS at Gly209, Gly251, and Gly265, however, coxsackievirus B3, a close relative of EV71, has been reported to cleave MAVS around Gln148 [20]. To test whether other sites could also be cleaved by EV71, we generated full-length RFP-NLS-MAVS as well as more truncations, including MAVS1 (aa 101–200/509–540) and MAVS3 (aa273–461/509–540) in addition to MAVS2 and MAVS4 as described above. Notably, all constructs contain the transmembrane domain to retain their mitochondria localization (Figure 2A). Although the full-length MAVS construct was correctly made, stable expression was not successful even after repeated attempt. When cells stably expressing MAVS1–4 were infected with EV71, we found that only MAVS2 could be cleaved by viral infection and RFP was translocated, while MAVS1, 3 and 4 was not (Figure 2B). These results suggest that no other cleavage sites exist, which is consistent with a previous report [22]. We then infected these different cells with Echo7 or CVB5, both of which are close members of the enterovirus genus. We found that, similar to EV71, both viruses cleaved MAVS2 but not MAVS1, 3 or 4 (Figure 2B).

These results indicated that MAVS cleavage is a conserved feature of enterovirus infection. These results were further corroborated by immunoblotting. We first established stable cell lines expressing different MAVS tagged with GFP (Figure 3A) and then infected these cells with different enteroviruses. Figure 3B showed that only MAVS 2 and full-length MAVS could be cleaved after viral infection, although all cells could be successfully infected. These results combined suggested that RFP-NLS-MAVS reporter could be cleaved by different enteroviruses and also indicated that all three enteroviruses that we tested could cleave MAVS at the same sites.

### 3.3. EV71, Echo7, and CVB5 Cleave the RFP-NLS-MAVS Reporter at the Same Residues

The fact that all three enteroviruses cleave the same MAVS truncation prompts us to speculate that they all cleave MAVS at the same sites. Sequence alignment of 2A proteins from three enteroviruses revealed that the sequences are conserved and the residues of the protease active site are highly conserved, i.e., C110, H21, and D39 (Figure 4A).

To test the above hypothesis, we constructed MAVS with single mutation (G209A, G251A or G265A) or triple mutations (G209A/G251A/G265A) at the cleavage sites and generated stable cell lines expressing these mutants (Appendix A). All mutants still maintained their mitochondria localization before infection, suggesting they are suitable to indicate viral infection by monitoring RFP translocation. In contrast to wild type MAVS2, RFP translocation was completely abolished in the triple mutant when cells were infected with any of these three enteroviruses (Figure 4B). These results suggested that all three viruses cleave MAVS at the same residues. In addition, all three viruses displayed similar cleavage preference in terms of single residual site. All viruses could still cleave G209A and G265A mutant, although not as efficient as wild type MAVS2. On the other hand, G251A mutant showed resistance to cleavage (Figure 4B), suggesting G251 is the most important cleavage site. These results were further confirmed by immunoblotting which showed that triple mutant and G251A mutant could not be cleaved by any virus, while G209A and G265A mutant could be still cleaved (Appendix A). Taken together, these results suggested that all three virus cleave MAVS at the same residues and this reporter system could be used to test the MAVS cleavage site of other enteroviruses or to study enterovirus-host interactions.

### 3.4. Time-Lapse Live-Cell Imaging of EV 71 Infection

A major advantage of this fluorescent reporter system is that it could be used for real-time visualization of EV71 infection, which could give us better understanding of EV71 infection kinetics in live cells at single-cell level. For this purpose, we conducted time-lapse imaging of Huh 7 cells stably expressing RFP-NLS-MAVS2 during EV71 infection. As expected, no nuclear translocation of RFP was observed in mock-infected cells (Figure 5A, upper panel, Appendix A), whereas in cells infected with EV71, nuclear translocation of RFP could be detected as early as 10 h post infection and complete cleavage was observed by 15 h (Figure 5A, middle panel, Appendix A). On the other hand, cells expressing RFP-NLS-MAVS4 did not show any translocation during the period of observation (Figure 5A, lower panel). Compared to current methods of viral infection detection, like viral protein expression or MAVS cleavage by immunoblotting (Figure 5B), this method could reveal infection much earlier and more importantly, this provides a method that could monitor viral infection in live cells at single-cell level.

### 3.5. Validation of Fluorescence-Based Reporter System for Antiviral Drug Screening

We next sought to identify whether this system could be used to evaluate the antiviral effects of compounds against enterovirus infection. For this purpose, cells expressing RFP-NLS-MAVS2 were first infected with EV71 for 4 h, followed by guanidine hydrochloride (GuHCl) treatment for another 20 h (Figure 6A), which is a known inhibitor of picornaviruses infection [10]. This compound showed no cytotoxicity below 10 mM as determined by cellular ATP content (Appendix A). After cells were treated with serial-diluted drugs ranging from 0.001 mM to 10 mM, images were taken and cells with RFP nucleus localization was counted and plotted against drug concentration (Figure 6B,C). In parallel, cells were also treated and analyzed by immunoblotting or q-RT-PCR to evaluate the inhibition of this compound on viral infection (Figure 6D,E). All methods generated quite consistent results showing that 0.1 mM GuHCl showed little inhibition on viral infection, while 1mM compound is potent against infection. After calculating the IC_50_ of this compound, we found comparable IC_50_ values generated from this reporter system compared to that from q-RT-PCR results (0.404 mM and 0.425 mM, respectively). These results indicated that this reporter system could be used to assess the efficacy of anti-enterovirus, and with proper optimization, this could be developed into a novel, high-content drug screening system.

## 4. Discussion

Human enterovirus is a diverse group of viruses that can cause a variety of diseases, ranging from mild hand, foot and mouth disease (HFMD) to more severe myocarditis, acute flaccid myelitis, encephalitis etc. [26,27,28]. Although these viruses caused frequent outbreaks in the past few decades, especially in the Asia-Pacific region, no specific treatment or effective vaccine—except for poliovirus, is available. Therefore, more efforts are needed to elucidate the details of these viral infections to aid vaccine and drug development. 

At present, a variety of methods have been developed to detect enterovirus infection, including PCR-based method to detect viral nucleotides and antibody-based method to detect the expression of viral proteins [29,30]. However, most of these methods require lysis or fixation of the cells and therefore, hinders real-time monitoring of enterovirus infection in live cells. Genetically engineering of virus with reporter such as GFP is an option for live-cell imaging but it always suffers with attenuation of virus production [23,31]. On the other hand, chemically labeling of virus with fluorescent materials always comes with modification of viral surface and therefore interfering with viral-host interactions [32,33]. In this study, we constructed a cell-based fluorescent reporter system to visualize enterovirus infection by taking advantage of the feature of MAVS cleavage by viral protease. Viral infection leads to MAVS cleavage and combined with fluorescence relocalization, these systems are sensitive and convenient to indicate enterovirus infection in live cells. Similar fluorescent viral reporter systems were previously described in HCV [34] and has been widely employed to study HCV viral infection [35], viral-induced immune response [36], and to screen antiviral drugs [37,38], suggesting these systems are useful tools for viral-host interaction studies and could facilitate antiviral drug development.

Recently, another form of translocation-based viral protease sensor has been reported to detect SARS-CoV-2 propagation in host cells. In this system, expression of the papain-like protease PLpro lead to cleavage of a specific linker and release the endoplasmic reticulum-anchored fluorescent protein to overall cellular distribution [39]. Nucleus/cytosol fluorescence ratio was then calculated to indicate viral infection. Compared to this method, RFP nuclear translocation employed in this study with the introduction of NLS is clearly visible and circumvent the quantification of both nucleus and cytosol fluorescence signals and therefore, make it a more straightforward approach to determine viral infection. 

Of note, in contrast to HCV induced RFP relocation, which showed perfect colocalization with Hoechst [34], RFP still showed residual amount of signals in cytoplasm after enterovirus infection. This was probably due to incomplete cleavage of MAVS by enterovirus or incomplete nuclear import of this MAVS truncation, as many cargo properties could affect nuclear transport, including size, surface and mechanical properties [40]. Nevertheless, we were able to show that the fluorescence translocation was caused by enterovirus infection and this feature allows simple, sensitive distinction of individual infected cells in live or fixed samples.

With these systems, we were also able to demonstrate that EV71, Echo 7 and CVB5 all cleaved MAVS at the same residues at Gly209, Gly251, and Gly265. CVB3 has been reported to cleave MAVS at Gln148 [20], however the CVB5 we used here did not seem to cleave MAVS1, which contain this Gln148 site, suggesting different strains of enterovirus could target different sites for MAVS cleavage. By constructing reporters with different MAVS truncations, this reporter system could be adapted to test the cleavage sites of other enterovirus strains and therefore provides a useful tool to study virus-host interactions.

More importantly, this reporter system could enable real-time visualization of EV71 infection by monitoring fluorescence nuclear translocation in live cells, thus providing a simple method applicable to image viral infection in individual cell. In addition, this reporter system could detect much earlier viral infection compared to the widely-used antibody-based detection methods, and would be of use to monitor early events of viral infection. 

Finally, this reporter system could also be optimized to test antiviral activity of small compound and could generate IC_50_ value similar to that of q-RT-PCR, suggesting this system is very sensitive. With this sensitive fluorescent feature, it would be easy to develop high-content screening by combining multiple fluorescent parameters with automated fluorescence imaging.

## Figures and Tables

**Figure 1 viruses-15-01064-f001:**
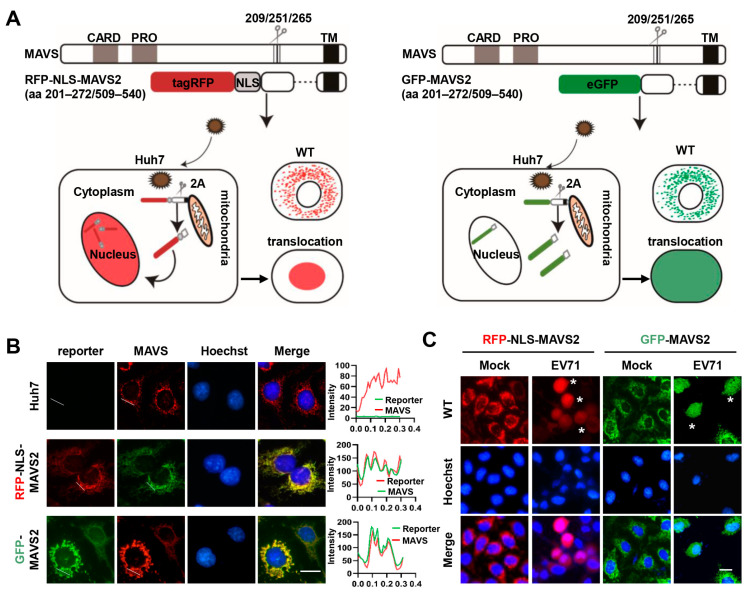
MAVS-based reporter systems for detection of EV71 infection. (**A**) Left: schematic diagram of RFP-NLS-MAVS2 reporter system. Right: schematic diagram of GFP-MAVS2 reporter system; (**B**) Huh 7 cells or cells stably expressing RFP-NLS-MAVS2 or GFP-MAVS2 were immunostained for MAVS expression with Hoechst nuclear counterstaining. Bar, 20 μm. The fluorescence intensity profiles along the selected lines are shown on the right; (**C**) Cells stably expressing RFP-NLS-MAVS2 or GFP-MAVS2 were infected with EV71 and fluorescence was monitored at 20 h post-infection. Nuclei were stained with Hoechst (blue). Asterisks indicate infected cells. Bar, 20 μm.

**Figure 2 viruses-15-01064-f002:**
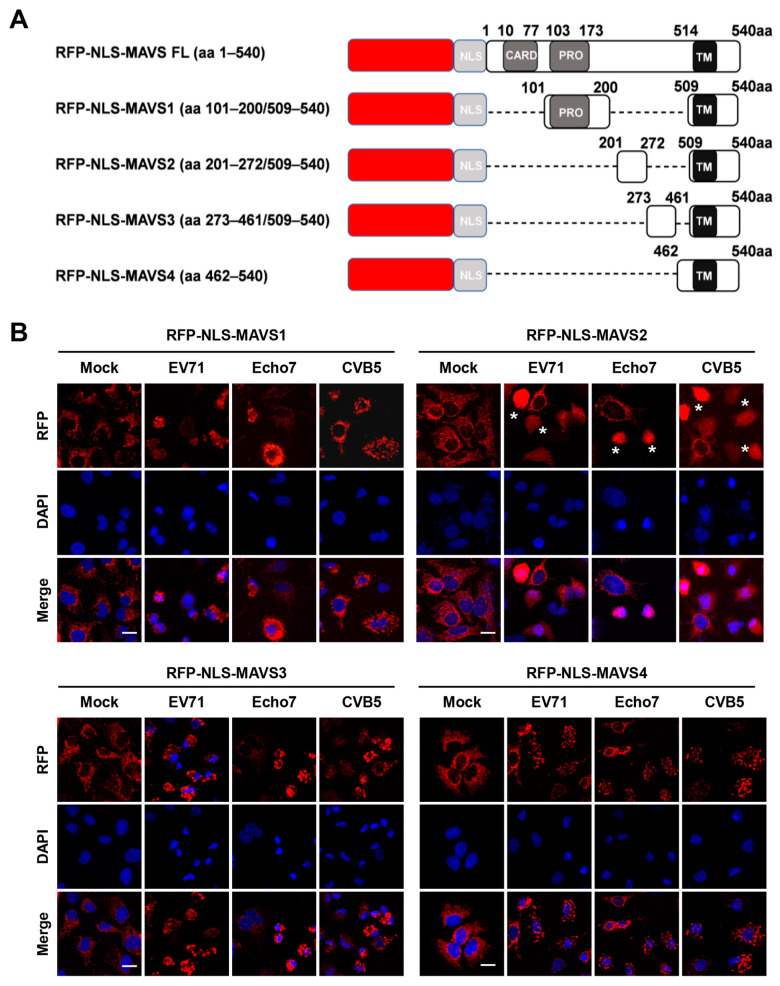
RFP-NLS-MAVS2 reporter could be cleaved by different enteroviruses. (**A**) Diagram of the RFP-NLS-MAVS and truncations. The locations of the CARD-like domain, proline-rich domain and transmembrane domain are indicated; (**B**) Huh 7 cells stably expressing the indicated reporter were infected with EV71 (MOI = 5), Echo7 (MOI = 5) or CVB5 (MOI = 2) and fluorescence was monitored at 20 h post-infection. Nuclei were stained with DAPI. Asterisks indicate infected cells. Bar, 20 μm.

**Figure 3 viruses-15-01064-f003:**
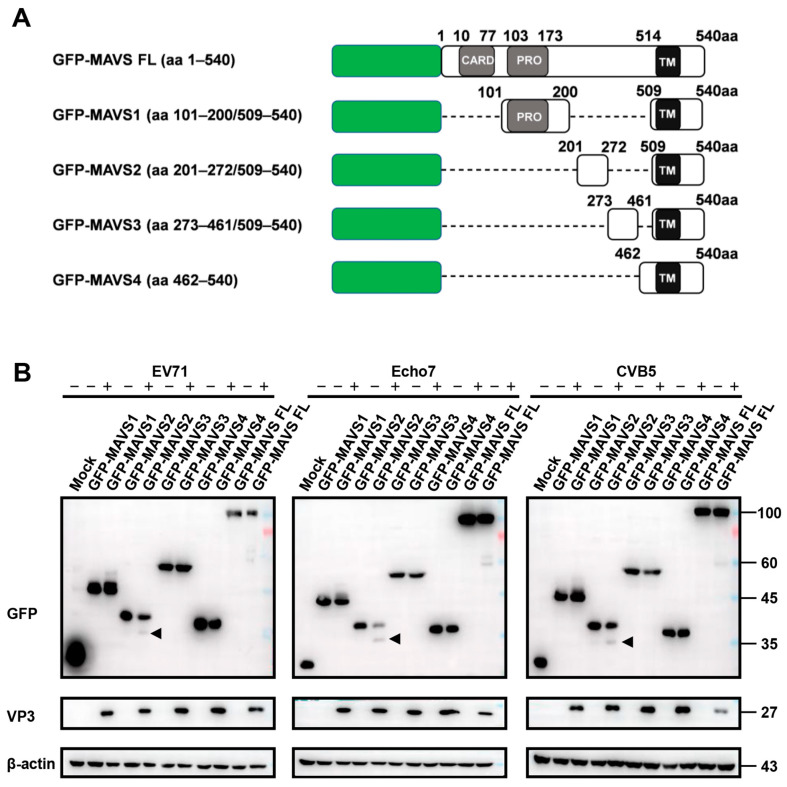
GFP-MAVS2 reporter could be cleaved by different enteroviruses. (**A**) Diagram of the GFP-MAVS and truncations; (**B**) Huh 7 cells stably expressing the indicated reporter were infected with EV71 (MOI = 5), Echo7 (MOI = 5) or CVB5 (MOI = 2) and GFP or VP3 expression was detected with immunoblotting. Arrowheads indicate cleaved MAVS. β-actin was shown as loading control.

**Figure 4 viruses-15-01064-f004:**
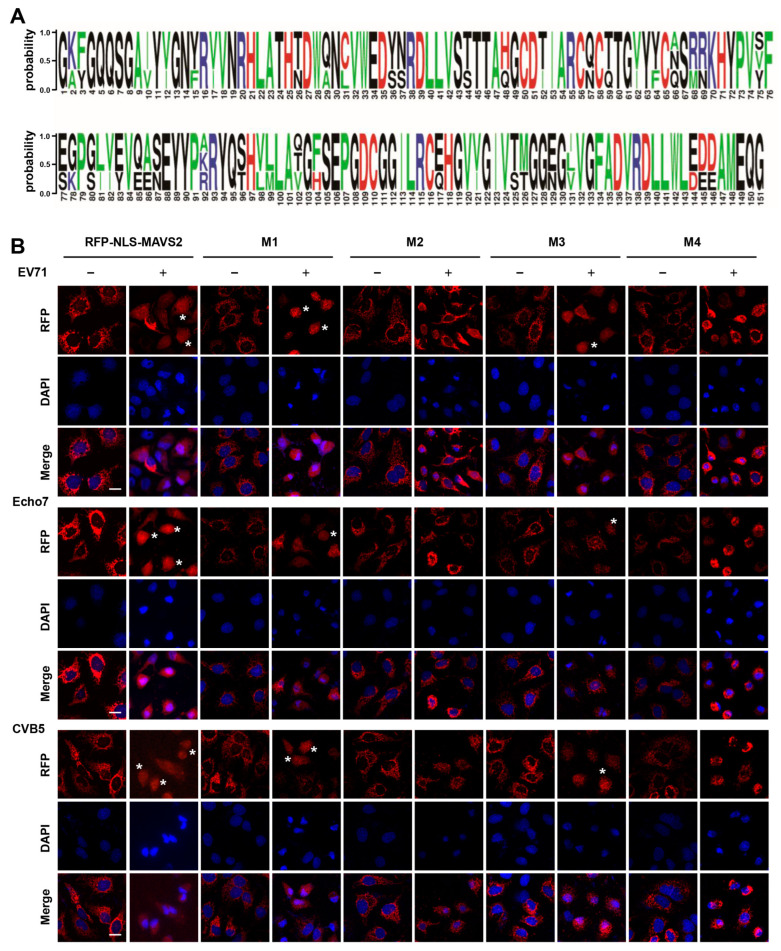
EV71, Echo7 and CVB5 cleave the RFP-NLS-MAVS reporter at the same residues. (**A**) Sequence alignment of 2A proteins from EV71, Echo7 and CVB5 with WebLogo 3. The letter height is proportional to the degree of amino acid conservation; (**B**) Huh 7 cells stably expressing the indicated reporter were infected with EV71, Echo7 or CVB5 and fluorescence was monitored at 24 h post-infection. Nuclei were stained with DAPI. Asterisks indicate infected cells. Bar, 20 μm.

**Figure 5 viruses-15-01064-f005:**
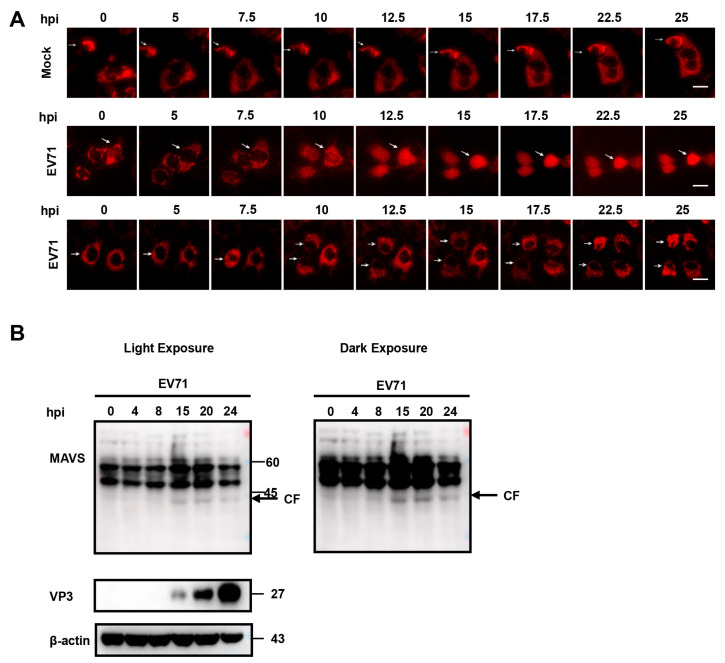
Time-lapse live-cell imaging of EV71 infection. (**A**) Huh 7 cells stably expressing RFP-NLS-MAVS2 (upper and middle panel) or RFP-NLS-MAVS4 (lower panel) were infected with EV71 (MO = 5) and fluorescent images were taken every 2.5 h. Bar, 20 μm; (**B**) Huh 7 cells stably expressing RFP-NLS-MAVS2 were infected with EV71 and cells were harvested at indicated hours after infection. Cell lysates were immunoblotted with MAVS antibody and VP3 antibody. The arrow indicates cleaved MAVS. β-actin was shown as loading control.

**Figure 6 viruses-15-01064-f006:**
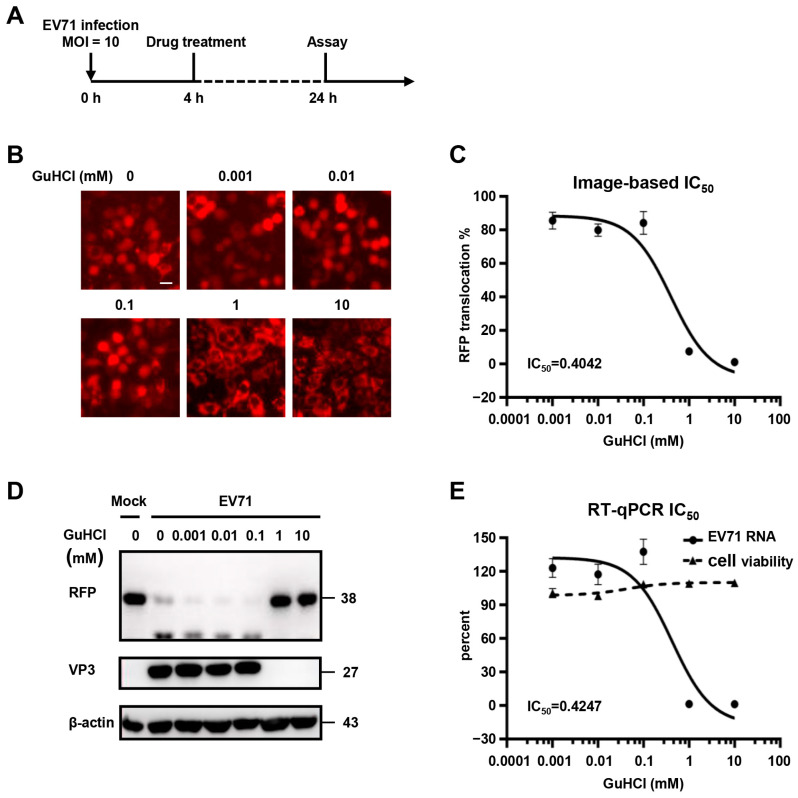
Validation of fluorescence-based reporter system for antiviral drug screening. (**A**) Schematic diagram of the experiment timeline; (**B**) Representative images of cells treated with indicated amount of GuHCl. Bar, 20 μm; (**C**) A dose-response curve of GuHCl treatment on RFP translocation; (**D**) Cells as described in (**A**) were immunoblotted with RFP or VP3 antibody. β-actin was shown as loading control; (**E**) Dose-response curves of GuHCl treatment on cell viability or viral RNA copy numbers. Cells as described in (**A**) were subjected to q-RT-PCR and viral RNA copy number were plotted against GuHCl concentration. IC_50_ were calculated with Prism. Cell viability was determined with Cell Titer Glo.

## Data Availability

The data presented in this study are available on request from the corresponding author.

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
