# Peer review of "MAVS-Based Reporter Systems for Real-Time Imaging of EV71 Infection and Antiviral Testing"

_viruses, 2023, doi:10.3390/v15051064_

Round 1
Reviewer 1 Report
This is an interesting and solid work on development and application of genetically encoded fluorescent sensor for the protease of EV71 and some other enteroviruses. The readout of the sensor is based on signal translocation from mitochondria surface to cytosol and nucleus. It was demonstrated to be a simple, clear and sensitive way to detect viral infection of cultured cells, either in real time or at the final point. It can be also used for testing activity of antiviral agents. I have minor concerns only:
1. Lines 147-149: “When these cells were infected with EV71 for 20 hrs, tagRFP was found to translocate into nucleus and colocalized with Hoechst, while GFP showed diffused distribution (Fig. 1C)…”
In fact, tagRFP after cleavage in most images clearly presents not only in nuclei but also in cytosol. This can be explained by either incomplete cleavage or incomplete nuclear import. This should be stated and discussed in the paper.
2. In Fig. 2C, in the panels for RFP-NLS-MASV2, CVB5 (right-most images) all cells look similarly, with red signal in the nuclei. Why?
3. Suppl. Fig 1C: Is there any confirmation of successful infection with HCV (like immunostaning in Suppl. Fig. 1A)?
4. I think it is more appropriate to place Table 1 (Primers used in this study) to the Supplementary rather than to the main manuscript.
5. Fig. 3B: What are the molecular weights of the colored marker bands?
6. In a recent paper, a similar translocation-based viral protease sensor was described: Sokolinskaya et al. Genetically Encoded Fluorescent Sensors for SARS-CoV-2 Papain-like Protease PLpro. Int J Mol Sci. 2022 Jul 15;23(14):7826. The authors may wish to note this.
7. Images in the main manuscript pdf and in Supplementary Movies are of low quality (probably because of file conversions). This should be improved if possible in the final version.
8. Line 265: a misprint “screenin”.
Reviewer 2 Report
In this manuscript, the authors present a reporter system for EV71 infection through the use of MAVS cleavage. Using two different constructs the authors show that EV71 infection leads to cleavage of the cytosolic portion of MAVS and release of a fluorescent protein from the mitochondria to relocate to either the nucleus via an NLS or diffuse cytosolic localization. The authors follow this up by making truncation mutants of MAVS with a TagRFP-NLS and show that only a certain region of MAVS is cleaved and relocalized during EV71 infection. Using similar constructs, the authors show that other enteroviruses, Echovirus 7 and CVB5, are able to cleave MAVS in the same region as EV71. They identified three potential cleavage sites within the region where MAVS was being cleaved by all three viruses' 2A protease. Then, they created mutations individually or in combination at these three residues.The authors found that mutations at individual residues only lead to one residue, G251, resistant to cleavage. Finally, the authors show that this reporter system can be applied to for antiviral drug testing by using a known inhibitor of picornaviruses, guanidine hydrochloride.
Overall, the experiments are well executed and the data largely support the conclusions. Two suggestions for improvement:
1. The authors should consider expressing the 2A protease and a dead protease independently of infection to show that 2A, and not 3C or a host protease, is directly involved in cleaving MAVS.
2. The naming of the MAVS residue mutants is a bit confusing since the authors previously named the truncation mutants 1-4. It would probably be better to name the residue mutants according to their specific mutation (e.g. G251A).
Minor comment: Please edit for grammar and clarity. For example, words like "residual" was used instead of "residue". These mistakes should be fixed.
Reviewer 3 Report
In the current study titled “MAVS-based reporter systems for real-time imaging of EV71 infection and antiviral testing”, the authors have developed fluorescent cell-based reporter systems that allow sensitive distinction of individual cells infected with EV71. The research is interesting to readers and represents a good scientific effort. Therefore, I recommend acceptance of the manuscript after minor revision.
1- Line 280, “anti-enterovirus inhibitors” the word inhibitors should be removed.
2- IC50 and CC50 should be corrected to IC50 and CC50.
3- Legends of the figures should be more concise. Instead, explain the figures in the main text.
4- Writing of the manuscript for language and grammar needs to be thoroughly checked.
